

# Frictional interactions between tidal constituents in tide-dominated estuaries

Huayang Cai[1], Marco Toffolon[2], Hubert H. G. Savenije[3], Qingshu Yang[1], and Erwan Garel[4]

[1]Institute of Estuarine and Coastal Research, School of Marine Sciences, Sun Yat-sen University, Guangzhou 510275, China

[2]Department of Civil, Environmental and Mechanical Engineering, University of Trento, Italy

[3]Department of Water Management, Faculty of Civil Engineering and Geosciences, Delft University of Technology, Netherlands

[4]Centre for Marine and Environmental Research (CIMA), University of Algarve, Faro, Portugal

*Correspondence to:* Erwan Garel (egarel@ualg.pt)

**Abstract.** When different tidal constituents propagate along an estuary, they interact because of the presence of nonlinear terms in the hydrodynamic equations. In particular, due to the quadratic velocity in the friction term, the effective friction experienced by both the predominant and the minor tidal constituents is enhanced. We explore the underlying mechanism with a simple conceptual

model by utilizing Chebyshev polynomials, enabling the effect of the velocities of the tidal constituents to be summed in the friction term and, hence, the linearized hydrodynamic equations to be solved analytically in a closed form. An analytical model is adopted for each single tidal constituent with a correction factor to adjust the linearized friction term, accounting for the mutual interactions between the different tidal constituents by means of an iterative procedure. The proposed method

is applied to the Guadiana (southern Portugal-Spain border) and the Guadalquivir (Spain) estuaries for different tidal constituents ($M_2$, $S_2$, $N_2$, $O_1$, $K_1$) imposed independently at the estuary mouth. The analytical results appear to agree very well with the observed tidal amplitudes and phases of the different tidal constituents.

## 1 Introduction

Numerous studies have been conducted in recent decades to model tidal wave propagation along an estuary since an understanding of tidal dynamics is essential for exploring the influence of human-induced (such as dredging for navigational channels) or natural (such as global sea level rises) interventions on estuarine environments (Schuttelaars et al., 2013; Winterwerp et al., 2013). Analytical models are invaluable tools and have been developed to study the basic physics of tidal dynamics




in estuaries; for instance, to examine the sensitivity of tidal properties (e.g., tidal damping or wave speed) to change in terms of external forcing (e.g., spring–neap variations of amplitude) and geometry (e.g., depth or channel length). However, most analytical solutions developed to date, which make use of the linearized Saint-Venant equations, can only deal with one predominant tidal constituent (e.g., $M_2$), which prevents consideration of the nonlinear interactions between different tidal

constituents. The underlying problem is that the friction term in the momentum equation follows a quadratic friction law, which causes a nonlinear behavior causing tidal asymmetry as tide propagates upstream. If the friction law were linear, one would expect that the effective frictional effect for different tidal constituents (e.g., $M_2$ and $S_2$) could be computed independently (Pingree, 1983).

To explore the interaction between different constituents of the tidal flow, the quadratic velocity

$u|u|$ (where $u$ is the velocity) is usually approximated by a truncated series expansion, such as a Fourier expansion (Proudman, 1953; Dronkers, 1964; Le Provost, 1973; Pingree, 1983; Fang, 1987; Inoue and Garrett, 2007). If the tidal current is composed of one dominant constituent and a much smaller second constituent, it has been shown by many researchers (Jeffreys, 1970; Heaps, 1978; Prandle, 1997) that the weaker constituent is acted on by up to 50% more friction than that of the

dominant constituent. However, this requires the assumption of a very small value of the ratio of the magnitudes of the weaker and dominant constituents, which indicates that this is only a first-order estimation. Later, some researchers have extended the analysis to improve the accuracy of estimates and to allow for more than two constituents (Pingree, 1983; Fang, 1987; Inoue and Garrett, 2007). Pingree (1983) investigated the interaction between $M_2$ and $S_2$ tides, resulting in a second-order

correction of the effective friction coefficient acting on the predominant $M_2$ tide and a fourth-order value for the weaker $S_2$ constituent of the tide. Fang (1987) derived exact expressions of the coefficients of the Fourier expansion of $u|u|$ for two tidal constituents but did not provide exact solutions for the case of three or more constituents. Later, Inoue and Garrett (2007) used a novel approach to determine the Fourier coefficients of $u|u|$, which allows the magnitude of the effective friction

coefficient to be determined for many tidal constituents. For the general two-dimensional tidal wave propagation, the expansion of quadratic bottom friction using a Fourier series was first proposed by Le Provost (1973) and subsequently applied to spectral models for regional tidal currents (Le Provost et al., 1981; Le Provost and Fornerino, 1985; Molines et al., 1989). Building on the previous work by Le Provost (1973), the importance of quadratic bottom friction in tidal propagation and damping was

discussed by Kabbaj and Le Provost (1980) and reviews of friction term in models were presented by Le Provost (1991).

In contrast, as noted by other researchers (Doodson, 1924; Dronkers, 1964; Godin, 1991, 1999), the quadratic velocity $u|u|$ is, mathematically, an odd function, and it is possible to approximate it by using a two- or three-term expression, such as $\alpha u + \beta u^3$ or $\alpha u + \beta u^3 + \xi u^5$, where $\alpha$, $\beta$, and $\xi$

are suitable numerical constants. The linear term $\alpha u$ represents the linear superposition of different constituents, while the nonlinear interaction is attributed to a cubic term $\beta u^3$ and a fifth-order term



$\xi u^5$. It is to be noted that such a method has the advantage of keeping the hydrodynamic equations resolvable in a closed form (Godin, 1991, 1999).

In this paper, a conceptual analytical model is presented to understand the propagation of different tidal constituents that one might wish to treat independently. The key lies in the treatment of the quadratic velocity in the friction term. The model has subsequently been applied to the Guadiana and the Guadalquivir estuaries in southern Iberian Peninsula, for which case the mutual interaction between the predominant $M_2$ tidal constituent and other tidal constituents (e.g., $S_2$, $N_2$, $O_1$, $K_1$) is explored.

## 2 Materials and methods

### 2.1 Hydrodynamic model

We are considering a semi-closed estuary that is forced by one predominant tidal constituent (e.g., $M_2$) with the tidal frequency $\omega = 2\pi/T$, where $T$ is the tidal period. As the tidal wave propagates into the estuary, it has a wave celerity of water level $c_A$, a wave celerity of velocity $c_V$, an amplitude of tidal elevation $\eta$, a tidal velocity amplitude $\upsilon$, a phase of water level $\phi_A$, and a phase of velocity $\phi_V$. The length of the estuary is indicated by $L_e$.

The geometry of a semi-closed estuary is shown in Figure 1, where $x$ is the longitudinal coordinate, which is positive in the landward direction, and $z$ is the free surface elevation. The tidally averaged cross-sectional area $\overline{A}$ and width $\overline{B}$ are assumed to be exponentially convergent in the landward direction, which can be described by

$$\overline{A} = \overline{A_0} \exp(-x/a), \tag{1}$$

$$\overline{B} = \overline{B_0} \exp(-x/b), \tag{2}$$

where $\overline{A_0}$ and $\overline{B_0}$ are the respective values at the estuary mouth (where $x=0$), and $a$ and $b$ are the convergence lengths of cross-sectional area and width, respectively. We also assume a rectangular cross-section, from which it follows that the tidally averaged depth is given by $\overline{h} = \overline{A}/\overline{B}$. The possible influence of storage area is described by the storage width ratio $r_S$, defined as the ratio of the storage width to the tidally averaged width (i.e., $r_S = B_S/\overline{B}$).

With the above assumptions, the one-dimensional continuity equation reads

$$r_S \frac{\partial h}{\partial t} + u \frac{\partial h}{\partial x} + h \frac{\partial u}{\partial x} + \frac{hu}{\overline{B}} \frac{d\overline{B}}{dx} = 0, \tag{3}$$

where $t$ is the time and $h$ the instantaneous depth. Assuming negligible density effects, the one-dimensional momentum equations can be cast as follows

$$\frac{\partial u}{\partial t} + u \frac{\partial u}{\partial x} + g \frac{\partial z}{\partial x} + \frac{g\,u|u|}{K^2 h^{4/3}} = 0, \tag{4}$$

where $g$ is the acceleration due to gravity and $K$ is the Manning-Strickler friction coefficient.




In order to obtain an analytical solution, we assume that the tidal amplitude is small with respect
to the mean depth and follow Toffolon and Savenije (2011) to derive the linearized solution of the
system of Eqs. (3) and (4). However, different from the standard linear solutions, we will retain
the mutual interaction among different harmonics originated by the nonlinear frictional term, which
contains two sources of nonlinearity: the quadratic velocity $u|u|$ and the variable depth at the denom-
inator. While we neglect the latter factor, consistent with the assumption of small tidal amplitude,
we will exploit Chebyshev polynomials to represent the harmonic interaction in the quadratic ve-
locity (see Section 3.1). For sake of clarity, we report here the linearized version of the momentum
equation

$$\frac{\partial u}{\partial t} + g\frac{\partial z}{\partial x} + \kappa\, u|u| = 0, \tag{5}$$

and the friction coefficient

$$\kappa = \frac{g}{K^2 \overline{h}^{4/3}}. \tag{6}$$

Toffolon and Savenije (2011) demonstrated that the tidal hydrodynamics in a semi-closed estuary
are controlled by a few dimensionless parameters that depend on geometry and external forcing
(for detailed information about analytical solutions for tidal hydrodynamics, readers can refer to
Appendix A). These parameters are defined in Table 1 and can be interpreted as follows.

The independent dimensionless parameters are: $\zeta_0$ is the dimensionless tidal amplitude (the sub-
script 0 indicating the seaward boundary condition); $\gamma$ is the estuary shape number (representing
the effect of cross-sectional area convergence); $\chi_0$ is the friction number (describing the role of the
frictional dissipation); $L_e^*$ is the dimensionless estuary length. The dimensional quantities used in
the definition of the dimensionless parameters are: $\eta_0$ is the tidal amplitude at the seaward boundary;
$c_0 = \sqrt{g\overline{h}/r_S}$ is the frictionless wave celerity in a prismatic channel; $L_0 = c_0/\omega$ is the tidal length
scale related to the frictionless tidal wave length by a factor $2\pi$.

The main dependent dimensionless parameters are also presented in Table 1, including: $\zeta$ is the
actual tidal amplitude; $\chi$ is the actual friction number; $\mu$ is the velocity number (the ratio of the actual
velocity amplitude to the frictionless value in a prismatic channel); $\lambda_A$ and $\lambda_V$ are, respectively,
the celerity for elevation and velocity (the ratio between the frictionless wave celerity in a prismatic
channel and actual wave celerity); $\delta_A$ and $\delta_V$ are, respectively, the amplification number for elevation
and velocity (describing the rate of increase, $\delta_A$ (or $\delta_V$) > 0, or decrease, $\delta_A$ (or $\delta_V$) < 0, of the
wave amplitudes along the estuary axis); $\phi = \phi_V - \phi_A$ is the phase difference between the phases of
velocity and elevation.

It is important to remark that several nonlinear terms are present both in the continuity and in the
momentum equations (Parker, 1991), which are responsible, for instance, of the internal generation
of overtides (e.g., $M_4$). In this approximated approach, we disregard them and focus exclusively
on the mutual interaction among the external tidal constituents mediated by the quadratic velocity



dependence in the frictional term. In fact, it crucially affects the propagation of the tidal waves
associated with the different constituents that are already present in the tidal forcing at the estuary
mouth.

## 2.2   Study areas

Both the Guadiana and the Guadalquivir estuaries are located in the southwest part of the Iberian
Peninsula. These systems are good candidates for the application of a 1D hydrodynamic model
of tidal propagation. Both estuaries are featured with a simple geometry, consisting of a single,
narrow and moderately deep channel with relatively smooth bathymetric variations. Moreover, their
tidal prism exceeds their average freshwater inputs by several orders of magnitude due to strong
regulation by dams. Under these largely predominant low river discharge conditions, both estuaries
are well-mixed, and the water circulation is mainly driven by tides.

The Guadiana estuary, at the southern border between Spain and Portugal, connects the Guadiana
River to the Gulf of Cadiz. Tidal water level oscillations are observed along the channel until a weir
located 78 km upstream of the river mouth (Garel et al., 2009). Both the cross-sectional area and the
channel width are convergent and can be described by an exponential function, with convergence
lengths of $a$=31 km and $b$=38 km, respectively (Figure 2). The flow depth is generally between 4 m
and 8 m, with a mean depth of about 5.5 m (Garel, 2017).

The tidal dynamics in the Guadiana estuary are derived from records obtained using eight pressure
transducers deployed for a period of 2 months (31 July to 25 September 2015) approximately every
10 km along the estuary (from the mouth to $\sim$ 70 km upstream). For each station, the amplitude
and phase of elevation of the tidal constituents were obtained from standard harmonic analysis of
the observed pressure records using the "t-tide" Matlab toolbox (Pawlowicz et al., 2002). The har-
monic results are displayed in Table 2. Near the mouth, the largest diurnal ($K_1$), semi-diurnal ($M_2$)
and quarter-diurnal ($M_4$) frequencies are similar to those previously reported at the same location
based on pressure records taken over $\sim$ 9 months (see Garel and Ferreira, 2013). In particular, the
value $(\eta_{K_1} + \eta_{O_1})/(\eta_{M_2} + \eta_{S_2})$ is less than 0.1 at the sea boundary, which indicates that the tide is
dominantly semi-diurnal.

The Guadalquivir estuary is located in southern Spain, at $\sim$ 100 km to the east of the Guadiana
River mouth. The estuary has a length of 103 km starting from the mouth at Sanlucar de Barrameda
to the Alcala del Rio dam. The geometry of the Guadalquivir estuary can be approximated by
exponential functions with convergence length of $a$=60 km for the cross-sectional area and $b$=66 km
for the width (see Diez-Minguito et al., 2012). The flow depth is more or less constant (7.1 m).

Tidal dynamics along the Gualdalquivir estuary was analysed by Diez-Minguito et al. (2012)
based on harmonic analyses of field measurements collected from June to December 2008. The
amplitude and phase of tidal constituents near the mouth are highly similar to those at the entrance
of the Guadiana estuary (Table 2), producing a semi-diurnal and mesotidal signal with a mean spring





tidal range of 3.5 m. In this paper, the tidal observations of the Guadalquivir estuary are directly taken from Diez-Minguito et al. (2012).

## 3  Conceptual model

### 3.1  Representation of quadratic velocity $u|u|$ using Chebyshev polynomials approach

The Chebyshev polynomials can be used to approximate the quadratic dependence of the friction term on the velocity, $u|u|$. Adopting a two-term approximation, it is known that (Godin, 1991, 1999)

$$u|u| = \widehat{v}^2 \left[ \alpha \left( \frac{u}{\widehat{v}} \right) + \beta \left( \frac{u}{\widehat{v}} \right)^3 \right], \tag{7}$$

where $\widehat{v}$ is the sum of the amplitudes of all the harmonic constituents. The Chebyshev coefficients

were determined as $\alpha = 16/(15\pi)$, and $\beta = 32/(15\pi)$ (Godin, 1991, 1999). It is important to note that, unlike series developments (e.g., Fourier expansion), the Chebyshev coefficients $\alpha$ and $\beta$ vary with the number of terms that are used in the development. Godin (1991) already showed that a two-term approximation (such as Eq. 7) is adequate to satisfactorily account for the friction.

For a single harmonic

$$u = v_1 \cos(\omega_1 t), \tag{8}$$

where $v_1$ is the velocity amplitude and $\omega_1$ its frequency, Eq. (7) can be expressed by exploiting standard trigonometric relations as

$$u|u| \cong v_1^2 \left[ \frac{8}{3\pi} \cos(\omega_1 t) + \frac{8}{15\pi} \cos(3\omega_1 t) \right]. \tag{9}$$

Focusing only on the original harmonic constituent leads to

$$u|u| \cong \frac{8}{3\pi} v_1^2 \cos(\omega_1 t), \tag{10}$$

which coincides exactly with Lorentz's classical linearization (Lorentz, 1926) or a Fourier expansion of $u|u|$ (Proudman, 1953).

Considering a second tidal constituent, the velocity is given by

$$u = v_1 \cos(\omega_1 t) + v_2 \cos(\omega_2 t) = \widehat{v} \left[ \varepsilon_1 \cos(\omega_1 t) + \varepsilon_2 \cos(\omega_2 t) \right], \tag{11}$$

where $v_2$ and $\omega_2$ are the amplitude and frequency of the second constituent, $\varepsilon_1 = v_1/\widehat{v}$ and $\varepsilon_2 = v_2/\widehat{v}$ are the ratios of the amplitudes to that of the maximum possible velocity $\widehat{v} = v_1 + v_2$. Note that the possible phase lag between the two constituents is neglected assuming a suitable time shift (Inoue and Garrett, 2007). In this case, the truncated Chebyshev polynomials approximation of $u|u|$ (focusing on two original tidal constituents) is expressed as (see also Godin, 1999)

$$u|u| \cong \frac{8}{3\pi} \widehat{v}^2 \left[ F_1 \varepsilon_1 \cos(\omega_1 t) + F_2 \varepsilon_2 \cos(\omega_2 t) \right], \tag{12}$$




with

$$F_1 = \frac{3\pi}{8}\left[\alpha + \beta\left(\frac{3}{4}\varepsilon_1^2 + \frac{3}{2}\varepsilon_2^2\right)\right] = \frac{1}{5}(2 + 3\varepsilon_1^2 + 6\varepsilon_2^2) = \frac{1}{5}\left(8 + 9\varepsilon_1^2 - 12\varepsilon_1\right), \tag{13}$$

$$F_2 = \frac{3\pi}{8}\left[\alpha + \beta\left(\frac{3}{4}\varepsilon_2^2 + \frac{3}{2}\varepsilon_1^2\right)\right] = \frac{1}{5}(2 + 3\varepsilon_2^2 + 6\varepsilon_1^2) = \frac{1}{5}\left(5 + 9\varepsilon_1^2 - 6\varepsilon_1\right), \tag{14}$$

where $F_1$ and $F_2$ represent the effective friction coefficients caused by the nonlinear interactions between tidal constituents. The last equality in Eqs. (13) and (14) is due to the fact that $\varepsilon_1 + \varepsilon_2 = 1$.

For illustration, approximations using Eqs. (7) and (12) for a typical tidal current with $\varepsilon_1 = 3/4$ and $\varepsilon_2 = 1/4$ are displayed in Figure 3 for the case of two tidal constituents. It can be seen that the Chebyshev polynomials approximation (Eq. 7) matches the nonlinear quadratic velocity well, while
Eq. (12), retaining only the original frequencies ($\omega_1$ and $\omega_2$), is still able to approximately capture the first-order trend of the quadratic term.

It can be seen from Eqs. (13) and (14) that when $\varepsilon_2 \ll 1$ (hence, $\varepsilon_1 \simeq 1$ for the dominant tidal constituent), $F_1 \simeq 1$, $F_2 \simeq 1.6$, thus the weaker constituent experiences proportionately 60% more friction than the dominant constituent, which is slightly larger than the classical result of 50% more
friction for the weaker tidal constituent. Figure 4 shows the solutions of effective friction coefficients $F_1$ and $F_2$ as a function of $\varepsilon_1$ for the case of two constituents. As expected, we see a symmetric response of these coefficients in the function of $\varepsilon_1$ since $\varepsilon_1 + \varepsilon_2 = 1$. Specifically, we note that the effective friction coefficient $F_1$ reaches a minimum when $\varepsilon_1=2/3$, when the velocity amplitude of the dominant constituent is twice larger than the weaker constituent.

Similarly, we are able to extend the same approach to the case of a generic number $n$ of astronomical tidal constituents (e.g., $K_1$, $O_1$, $M_2$, $S_2$, $N_2$)

$$u = \sum_{i=1}^{n} v_1 \cos(\omega_i t) = \widehat{v}\sum_{i=1}^{n} \varepsilon_i \cos(\omega_i t), \tag{15}$$

in which the subscript $i$ represents the $i$-th tidal constituent. Considering only the original tidal constituents, the quadratic velocity can be approximated as

$$u|u| \cong \frac{8}{3\pi}\widehat{v}^2 \sum_{i=1}^{n} F_i \varepsilon_i \cos(\omega_i t), \tag{16}$$

and the general expression for the effective friction coefficients of $j$-th tidal constituents is given by

$$F_j = \frac{3\pi}{8}\left\{\alpha + \beta\left[\sum_{i=1,i\neq j}^{n}\frac{3}{2}\varepsilon_i^2 - \frac{3}{4}\varepsilon_j^2\right]\right\} = \frac{1}{5}\left(2 + 3\varepsilon_j^2 + \sum_{i=1,i\neq j}^{n} 6\varepsilon_i^2\right). \tag{17}$$

We provide the complete coefficients for the cases of one to three constituents in Appendix B.

### 3.2 Effective friction in the momentum equation

For a single tidal constituent $u = v_1\cos(\omega_1 t)$, the quadratic velocity term $u|u|$ is often approximated by adopting Lorentz's linearization equation (Eq. 10) and thus the friction term in Eq. (5) becomes

$$\kappa u|u| = \left(\kappa\frac{8}{3\pi}v_1\right)u = ru, \tag{18}$$





which is the "standard" case for a monochromatic wave, i.e. when we only deal with a predominant tidal constituent (e.g., $M_2$).

For illustration of the method, we consider a tidal current that is composed of one dominant constituent (e.g., $M_2$ with velocity $u_1$) and a weaker constituent (e.g., $S_2$ with velocity $u_2$), which is a simple but important example in estuaries, i.e., $u = u_1 + u_2$. In this case, combination of Eq. (5) and the Chebyshev polynomials expansion of $u|u|$ (Eq. 12) yields

$$\frac{\partial u_1}{\partial t} + \frac{\partial u_2}{\partial t} + g\frac{\partial z_1}{\partial x} + g\frac{\partial z_2}{\partial x} + \kappa\frac{8}{3\pi}\widehat{\upsilon}(F_1 u_1 + F_2 u_2) = 0 , \tag{19}$$

where $z_1$ is the free surface elevation for the dominant constituent and $z_2$ for the secondary constituent. Exploiting the linearity of Eq. (19), we can solve the two problems independently. As a result, we see that the actual friction term that is felt in Eq. (19) is different from that would be felt by the single constituent alone (Eq. 18).

Introducing a general form of the linearized momentum equation for the generic $i$-th constituent

$$\frac{\partial u_i}{\partial t} + g\frac{\partial z_i}{\partial x} + f_i r_i u_i = 0 , \tag{20}$$

with

$$r_i = \kappa\frac{8}{3\pi}\upsilon_i , \tag{21}$$

as in the standard case, we see that the effective friction term contains a correction factor

$$f_i = \frac{F_i}{\varepsilon_i} , \tag{22}$$

through the coefficient $F_i$. Since the ratio $\varepsilon_i$ can be quite small for a weaker constituent, the friction actually felt can be significantly stronger.

## 4 Results

### 4.1 Hydrodynamic modeling incorporating the friction correction factor

If there are many tidal constituents, then the friction experienced by one is affected by the others. As suggested by our conceptual model, the mutual effects can be incorporated by using the friction correction factor $f_n$ defined in Eq. (22) if the other (weaker) constituents are treated in the same way as the predominant constituent. As a result, the friction number $\chi_n$ for each tidal constituent can be modified as

$$\chi_n = f_n\chi , \tag{23}$$

where $\chi$ is the friction number (see definition in Table 1) experienced if only a single tidal constituent is considered.





We note that the modified friction number $\chi_n$ in Eq. (23) contains the friction coefficient $K$. In many applications, $K$ is calibrated separately for each tidal constituent to account for the different friction exerted due to the combined tide, either changing $K$ directly or through calibration of the different correction friction factors $f_n$ (see, e.g., Cai et al., 2015, 2016). The current study aims at avoiding the need to adjust $K$ individually, so that only a single value of $K$ can be calibrated, which is based on the physical consideration that friction mostly depends on bottom roughness, and the other factors (tide interaction) are to be correctly modelled.

### 4.2 Procedure to study the propagation of the different constituents

With a hydrodynamic model for a single constituent (see Appendix A), an iterative procedure can be designed to study the propagation of the different constituents by calibrating a single value of the Manning-Strickler friction parameter $K$. The flow chart illustrating the computation process is presented in Figure 5. Initially, we assume the friction correction factor $f_i=1$ for each tidal constituent, and compute the first tentative values of velocity amplitude $\upsilon_i$ along the channel using the hydrodynamic model. This allows defining $\widehat{\upsilon}$ and, hence, $\varepsilon_i$. Taking into account the frictional interaction between tidal constituents, the revised $f_i$ is calculated using Eqs. (17) and (22). Subsequently, using the updated $f_i$, the new velocity amplitude $\upsilon_i$ along the channel can be computed using the hydrodynamic model. This process is repeated until the result is stable. In this paper, two examples of Matlab scripts are provided together with the observed tidal data in the Guadiana and Guadalquivir estuaries (see Supporting Information).

It is worth stressing that the single constituents are not calibrated independently, as was done in previous analyses (e.g., Cai et al., 2015). Conversely, only a single friction parameter, $K$, is calibrated or estimated based on the physical knowledge of the system (bed roughness). This feature represents a major advantage of the proposed method because the frictional interaction is modelled in mechanistic terms using Eq. (22).

### 4.3 Application to the Guadiana and Guadalquivir estuaries

In this study, the analytical model for a semi-closed estuary presented in Section 2.1 was applied to the Guadiana and Guadalquivir estuaries to reproduce the correct tidal behavior for different tidal constituents. The analytical results were compared with observed tidal amplitude $\eta$ and associated phase of elevation $\phi_A$.

The morphology of the Guadiana estuary was represented in the model with a constant depth (5.5 m), an exponentially converging width (length scale, 38 km) and a constant storage ratio of 1 representative of the limited salt marsh areas (about 20 km$^2$, see Garel (2017)). The Manning-Strickler friction coefficient ($K$ = 42 m$^{1/3}$s$^{-1}$) was determined by calibrating the model outputs (obtained using the iterative procedure presented in section 4.2) with observations. It can be seen from Figure 6 that the computed tidal amplitude and phase of elevation are in good agreement with



the observed values for different tidal constituents in the Guadiana estuary. The $N_2$ amplitude is
slightly overestimated in the central part of the estuary, which may suggest that the harmonic analysis
has some difficulties to resolve this constituent in relation to the length of the considered time series
290   (54 days). In support, the $N_2$ amplitude (0.16 m) from a longer time series (85 days) collected in
2017 at 58 km from the mouth matches better the model output, while results for other constituents
are similar in 2015 and 2017 (Garel, unpublished data). Otherwise, the correspondence is poorest for
the semi-diurnal constituents at the most upstream station, owing to truncation of the lowest water
levels by a sill located at about 65 km from the river mouth (Garel, 2017). Table 3 displays the mean
friction correction coefficient $f$ obtained from the iterative procedure to account for the nonlinear
interaction between different tidal constituents. In particular, the mean friction correction factors $f$
for the minor constituents $S_2$, $N_2$, $O_1$, and $K_1$ are 4.6, 8.1, 41.1, and 49.8, respectively.

To understand the tidal dynamics between different tidal constituents along the Guadiana estuary,
the longitudinal variations of the tidal damping/amplification number $\delta_A$ and celerity number $\lambda_A$ (see
their definitions in Table 1) are shown in Figure 7 where similar minor constituents in semidiurnal
($S_2$, $N_2$) and diurnal ($O_1$, $K_1$) band behave more or less the same. As shown in Figure 7a, the
minor constituents $S_2$, $N_2$, $O_1$, and $K_1$ experience more friction compared with the predominant
$M_2$ tide. Interestingly, we observe a stronger damping ($\delta_A < 0$) of semidiurnal constituents ($S_2$,
$N_2$) than those of diurnal constituents ($O_1$, $K_1$) in the seaward part of the estuary (around $x$=0-40
km) although the amplitudes of the diurnal constituents are less than those of the semidiurnal ones.
In contrast, the amplification ($\delta_A > 0$) of semidiurnal constituents ($S_2$, $N_2$) is more apparent than
those of diurnal constituents ($O_1$, $K_1$) in the landward part of the estuary. For the wave celerity, as
expected the dominant $M_2$ tide travels faster (smaller $\lambda_A$) than minor tidal constituents. In addition,
we observe that the wave celerity of semidiurnal tidal constituents is larger than those of diurnal
constituents in the seaward reach (around $x$=0-30 km), while it is the opposite in the landward reach,
which suggests a complex relation between tidal damping/amplification and wave celerity due to the
combined impacts of channel convergence, bottom friction and reflected wave.

For the Guadalquivir estuary, the geometry can be approximated as a converging estuary with
a width convergence length of $b$=65.5 km and a constant stream depth of about 7.1 m. A linear
reduction of the storage width ratio of 1.5-1 was adopted over the reach 0-103 km. The observed
tidal amplitudes and phases are best reproduced by using the model for $K = 46 \, \mathrm{m}^{1/3}\mathrm{s}^{-1}$ (see Figure
8). In general, the observed tidal properties (tidal amplitude and phase) of different constituents are
well reproduced. The enhanced frictional coefficient f for minor constituents $S_2$, $N_2$, $O_1$, and $K_1$
are 5.4, 9.7, 40.7, and 43.7, respectively (Table 3).
Figure 9 shows the longitudinal variations of tidal damping/amplification and wave celerity for
the Guadalquivir estuary. Similar to the Guadiana estuary, we observe that the dominant $M_2$ tide
experiences less tidal damping and travels faster than other minor tidal constituents. It can be seen
from Figure 9 that the magnitude of tidal damping is approximately one order larger than that in





the Guadiana estuary (Figure 9a) and hence the wave celerity is comparatively smaller (larger $\lambda_A$,

Figure 9b). Unlike the Guadiana estuary, the damping experienced by the minor semidiurnal tides is less than those of diurnal constituents in the seaward reach (around $x$=0-55 km), while the wave celerity is consistently larger for the whole channel.

## 5 Conclusions

In this study, we provide insight into the mutual interactions between one predominant (e.g., $M_2$)

and other tidal constituents in estuaries and the role of quadratic friction on tidal wave propagation. An analytical method exploiting the Chebyshev polynomials was developed to quantify the effective friction experienced by different tidal constituents. Based on the linearization of the quadratic friction, the conceptual model has been used to explore the nonlinear interaction of different tidal constituents, which enables them to be treated independently by means of an iterative procedure.

Thus, an analytical hydrodynamic model for a single tidal constituent can be used to reproduce the correct wave behavior for different tidal constituents. In particular, it was shown that a correction of the friction term needs to be used to correctly reproduce the tidal dynamics for minor tidal constituents. The application to the Guadiana and the Guadalquivir estuaries shows that the conceptual model can interpret the nonlinear interaction reasonably well when combined with an analytical

model for tidal hydrodynamics.

A crucial feature of the proposed approach is the deterministic description of the mutual frictional interaction among tidal constituents, which avoids the need of an independent calibration of the friction parameter for the single constituent. In this respect, further work is required to explore whether a reliable value of the friction coefficient estimated through this method can be parametrized

based on observations of the bottom roughness of the estuary.

## Appendix A

### Analytical solutions of tidal hydrodynamics for a single tidal constituent

In this paper, analytical solutions for a semi-closed estuary proposed by Toffolon and Savenije (2011) were used to reproduce the longitudinal tidal dynamics along the estuary axis. The solution makes

use of the parameters that are defined in Table 1.

The analytical solutions for the tidal wave amplitudes and phases are given by:

$$\eta = \zeta_0 \, \overline{h_0} \, |A^*|, \qquad \upsilon = r_S \, \zeta_0 \, c_0 \, |V^*|, \tag{A1}$$

$$\tan(\phi_A) = \frac{\Im(A^*)}{\Re(A^*)}, \qquad \tan(\phi_V) = \frac{\Im(V^*)}{\Re(V^*)}, \tag{A2}$$



where $\Re$ and $\Im$ are the real and image parts of the corresponding term, and $A^*$ and $V^*$ are unknown complex functions varying along the dimensionless coordinate $x^* = x/L_0$:

$$A^* = a_1^* \exp\left(w_1^* x^*\right) + a_2^* \exp\left(w_2^* x^*\right), \tag{A3}$$

$$V^* = v_1^* \exp\left(w_1^* x^*\right) + v_2^* \exp\left(w_2^* x^*\right). \tag{A4}$$

For a tidal channel with a closed end, the analytical solutions for the unknown variables in Eqs. (A3) and (A4) are listed in Table 4, where $\Lambda$ is a complex variable, defined as

$$\Lambda = \sqrt{\gamma^2/4 - 1 + i\widehat{\chi}}, \qquad \widehat{\chi} = \frac{8}{3\pi}\mu\chi, \tag{A5}$$

where the coefficient $8/(3\pi)$ stems from the adoption of Lorentz's linearization when considering only one single predominant tidal constituent (e.g., $M_2$).

Since the friction parameter $\widehat{\chi}$ depends on the unknown value of $\mu$ (or $\upsilon$), an iterative procedure was used to determine the correct wave behavior. In addition, to account for the longitudinal variation of the cross-section (e.g., estuary depth) a multi-reach technique was adopted by subdividing the entire estuary into multiple sub-reaches and the solutions obtained by solving a set of linear equations with internal boundary conditions at the junction of the sub-reaches satisfying the continuity condition (see details in Toffolon and Savenije, 2011).

For given computed values of $A^*$ and $V^*$, the dependent parameters defined in Table 1 can be computed using the following equations:

$$\mu = |V^*|, \qquad \phi = \phi_V - \phi_A, \tag{A6}$$

$$\delta_A = \Re\left(\frac{1}{A^*}\frac{\mathrm{d}A^*}{\mathrm{d}x^*}\right), \qquad \delta_V = \Re\left(\frac{1}{V^*}\frac{\mathrm{d}V^*}{\mathrm{d}x^*}\right), \tag{A7}$$

$$\lambda_A = \left|\Im\left(\frac{1}{A^*}\frac{\mathrm{d}A^*}{\mathrm{d}x^*}\right)\right|, \qquad \lambda_V = \left|\Im\left(\frac{1}{V^*}\frac{\mathrm{d}V^*}{\mathrm{d}x^*}\right)\right|. \tag{A8}$$

**Appendix B**

**Coefficients of the Godin's expansion**

The following trigonometric equation

$$\cos^3(\omega_1 t) = \frac{3}{4}\cos(\omega_1 t) + \frac{1}{4}\cos(3\omega_1 t), \tag{B1}$$

is used to convert the third-order terms of Eq. (7) to the harmonic constituents. For a single harmonic, it follows that





$$u|u| = v_1^2 \left[ \left( \alpha + \frac{3}{4}\beta \right) \cos(\omega_1 t) + \frac{1}{4}\beta \cos(3\omega_1 t) \right]. \tag{B2}$$

For two harmonic constituents, the Chebyshev polynomials approximation of $u|u|$ is expressed as

$$u|u| = v_1^2 \left\{ \alpha \left[ \varepsilon_1 \cos(\omega_1 t) + \varepsilon_2 \cos(\omega_2 t) \right] + \beta \left[ \varepsilon_1 \cos(\omega_1 t) + \varepsilon_2 \cos(\omega_2 t) \right]^3 \right\}. \tag{B3}$$

In Eq. (B3), the cubic term can be expanded as

$$\begin{aligned}
\left[ \varepsilon_1 \cos(\omega_1 t) + \varepsilon_2 \cos(\omega_2 t) \right]^3 = {}& \varepsilon_1^3 \cos^3(\omega_1 t) + 3\varepsilon_1 \varepsilon_2^2 \cos(\omega_1 t) \cos^2(\omega_2 t) \\
& + 3\varepsilon_2 \varepsilon_1^2 \cos(\omega_2 t) \cos^2(\omega_1 t) + \varepsilon_2^3 \cos^3(\omega_2 t).
\end{aligned} \tag{B4}$$

Making use of the trigonometric equations to expand the power of the cosine functions (e.g.,
$\cos^3(\omega_1 t)$ and $\cos^2(\omega_1 t)$) and extracting only the harmonic terms with frequencies $\omega_1$ and $\omega_2$, Eq.
(B3) can be reduced to Eq. (12).

For the case of many constituents, here we only provide the exact coefficients for $n=3$:

$$F_1 = \frac{3\pi}{8} \left[ \alpha + \beta \left( \frac{3}{4}\varepsilon_1^2 + \frac{3}{2}\varepsilon_2^2 + \frac{3}{2}\varepsilon_3^2 \right) \right] = \frac{1}{5} \left( 2 + 3\varepsilon_1^2 + 6\varepsilon_2^2 + 6\varepsilon_3^2 \right), \tag{B5}$$

$$F_2 = \frac{3\pi}{8} \left[ \alpha + \beta \left( \frac{3}{4}\varepsilon_2^2 + \frac{3}{2}\varepsilon_1^2 + \frac{3}{2}\varepsilon_3^2 \right) \right] = \frac{1}{5} \left( 2 + 3\varepsilon_2^2 + 6\varepsilon_1^2 + 6\varepsilon_3^2 \right), \tag{B6}$$

$$F_3 = \frac{3\pi}{8} \left[ \alpha + \beta \left( \frac{3}{4}\varepsilon_3^2 + \frac{3}{2}\varepsilon_1^2 + \frac{3}{2}\varepsilon_2^2 \right) \right] = \frac{1}{5} \left( 2 + 3\varepsilon_3^2 + 6\varepsilon_1^2 + 6\varepsilon_2^2 \right). \tag{B7}$$

Equations (B5) to (B6) reduce to Eqs. (13) and (14) when $\varepsilon_3 = 0$ (i.e., $v_3=0$).

*Acknowledgements.* We acknowledge the financial support from the National Key R&D of China (Grant No.
2016YFC0402600), from the National Natural Science Foundation of China (Grant No. 51709287), from the
Basic Research Program of Sun Yat-Sen University (Grant No. 17lgzd12) and from the Water Resource Science
and Technology Innovation Program of Guangdong Province (Grant No. 2016-20). The work of the fifth author
was supported by FCT research contract IF/00661/2014/CP1234.



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




**Table 1.** Definitions of dimensionless parameters.

| Independent parameters | Dependent parameters |
|---|---|
| Tidal amplitude at the mouth | Tidal amplitude |
| $\zeta_0 = \eta_0/\overline{h_0}$ | $\zeta = \eta/\overline{h}$ |
| Friction number at the mouth | Friction number |
| $\chi_0 = r_S c_0 \zeta_0 g / \left( K^2 \omega \overline{h_0}^{-4/3} \right)$ | $\chi = r_S c_0 \zeta g / \left( K^2 \omega \overline{h}^{-4/3} \right)$ |
| Estuary shape | Velocity number |
| $\gamma = c_0/(\omega a)$ | $\mu = \upsilon/(r_S \zeta c_0) = \upsilon \overline{h}/(r_S \eta c_0)$ |
| Estuary length | Damping number for water level |
| $L_e^* = L_e/L_0$ | $\delta_A = c_0 \mathrm{d}\eta/(\eta \omega \mathrm{d}x)$ |
|  | Damping number for velocity |
|  | $\delta_V = c_0 \mathrm{d}\upsilon/(\upsilon \omega \mathrm{d}x)$ |
|  | Celerity number for water level |
|  | $\lambda_A = c_0/c_A$ |
|  | Celerity number for velocity |
|  | $\lambda_V = c_0/c_V$ |
|  | Phase difference |
|  | $\phi = \phi_V - \phi_A$ |





**Table 2.** Tidal elevation amplitudes (m) and phases (°) estimates (with 95% confidence intervals in brackets) from harmonic analyses of pressure records along the Guadiana estuary ($x$: distance from the mouth, km).

| | Amplitude (m) | | | | | | | |
|---|---|---|---|---|---|---|---|---|
| $x$ (km) | $M_{sf}$ | $O_1$ | $K_1$ | $N_2$ | $M_2$ | $S_2$ | $M_4$ | $M_6$ |
| 2.4 | 0.01 (0.03) | 0.06 (0.01) | 0.07 (0.01) | 0.23 (0.01) | 0.97 (0.01) | 0.37 (0.02) | 0.02 (0.00) | 0.01 (0.00) |
| 10.7 | 0.01 (0.07) | 0.06 (0.01) | 0.07 (0.01) | 0.22 (0.01) | 0.93 (0.01) | 0.34 (0.01) | 0.02 (0.01) | 0.01 (0.00) |
| 22.8 | 0.03 (0.04) | 0.06 (0.01) | 0.07 (0.01) | 0.20 (0.02) | 0.86 (0.02) | 0.29 (0.02) | 0.04 (0.01) | 0.02 (0.01) |
| 33.9 | 0.06 (0.05) | 0.06 (0.01) | 0.07 (0.01) | 0.20 (0.02) | 0.85 (0.02) | 0.27 (0.02) | 0.04 (0.01) | 0.03 (0.01) |
| 43.6 | 0.06 (0.06) | 0.06 (0.01) | 0.07 (0.01) | 0.21 (0.02) | 0.87 (0.02) | 0.27 (0.02) | 0.05 (0.01) | 0.03 (0.01) |
| 51.4 | 0.05 (0.05) | 0.06 (0.01) | 0.07 (0.01) | 0.22 (0.02) | 0.90 (0.02) | 0.28 (0.02) | 0.07 (0.01) | 0.03 (0.01) |
| 60.1 | 0.07 (0.06) | 0.06 (0.01) | 0.07 (0.01) | 0.22 (0.02) | 0.93 (0.02) | 0.30 (0.02) | 0.08 (0.01) | 0.04 (0.01) |
| 69.6 | 0.10 (0.06) | 0.06 (0.01) | 0.06 (0.01) | 0.19 (0.03) | 0.78 (0.03) | 0.24 (0.03) | 0.16 (0.03) | 0.02 (0.01) |
| | Phase (°) | | | | | | | |
| 2.4 | 190 (149) | 310 (6) | 73 (5) | 54 (4) | 62 (1) | 93 (2) | 151 (8) | 219 (18) |
| 10.7 | 8 (190) | 319 (7) | 85 (6) | 68 (3) | 75 (1) | 108 (3) | 103 (14) | 237 (15) |
| 22.8 | 38 (66) | 331 (9) | 103 (7) | 87 (4) | 93 (1) | 130 (3) | 131 (12) | 294 (16) |
| 33.9 | 49 (56) | 343 (7) | 116 (6) | 104 (5) | 109 (1) | 151 (4) | 166 (8) | 336 (11) |
| 43.6 | 51 (58) | 348 (8) | 123 (8) | 116 (5) | 121 (1) | 166 (4) | 189 (6) | 12 (14) |
| 51.4 | 48 (48) | 352 (9) | 128 (8) | 123 (6) | 128 (1) | 175 (5) | 203 (5) | 43 (19) |
| 60.1 | 53 (58) | 356 (9) | 133 (8) | 131 (6) | 135 (1) | 184 (5) | 219 (4) | 69 (21) |
| 69.6 | 51 (43) | 7 (9) | 146 (8) | 146 (9) | 148 (2) | 200 (7) | 261 (11) | 15 (18) |

**Table 3.** Mean correction friction factor $f$ for different tidal constituents along the Guadiana and the Guadalquivir estuaries.

| Tidal constituents | $M_2$ | $S_2$ | $N_2$ | $K_1$ | $O_1$ |
|---|---|---|---|---|---|
| Guadiana | 1.1 | 4.6 | 8.1 | 41.1 | 49.8 |
| Guadalquivir | 1.1 | 5.4 | 9.7 | 40.7 | 43.7 |

**Table 4.** Analytical expressions for unknown complex variables for the case of a closed estuary.

| $a_1^*, a_2^*$ | $v_1^*, v_2^*$ | $w_1^*, w_2^*$ |
|---|---|---|
| $a_1^* = \left[1 + \exp\left(\Lambda L_e^*\right)\frac{\Lambda+\gamma/2}{\Lambda-\gamma/2}\right]^{-1}$ | $v_1^* = \frac{-i a_1^*}{\Lambda-\gamma/2}$ | $w_1^* = \gamma/2 + \Lambda$ |
| $a_2^* = 1 - a_1^*$ | $v_2^* = \frac{i(1-a_1^*)}{\Lambda+\gamma/2}$ | $w_2^* = \gamma/2 - \Lambda$ |



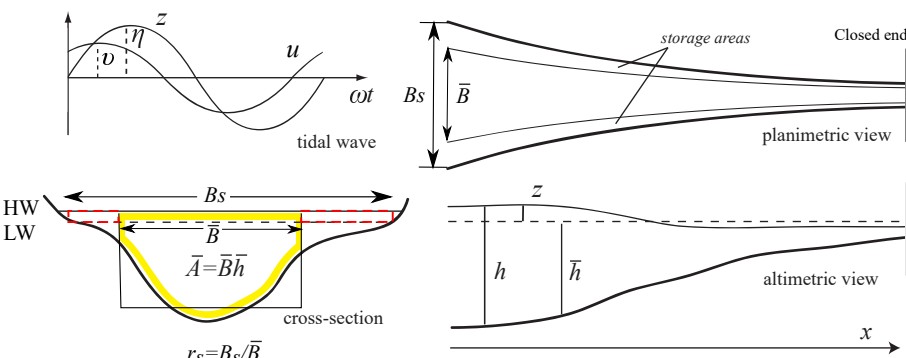

**Figure 1.** Geometry of a semi-closed estuary and basic notation (after Savenije et al. (2008)). HW, high water; LW, low water.

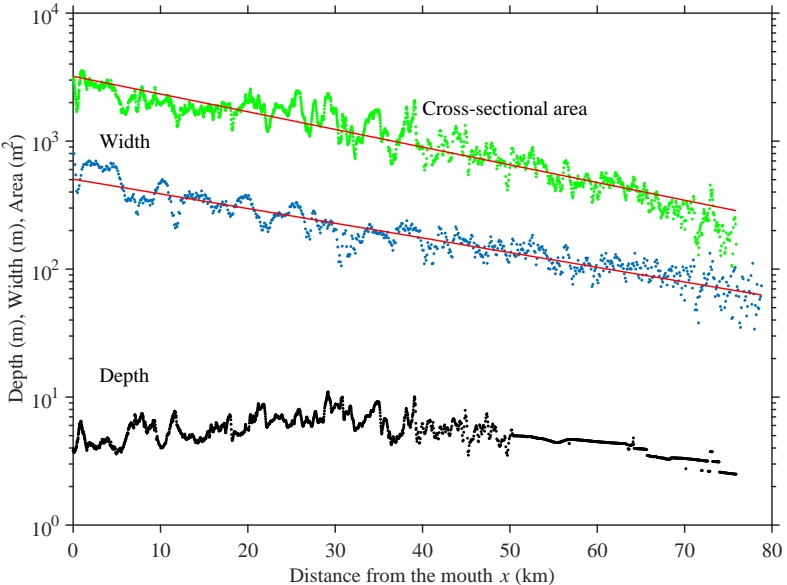

**Figure 2.** Tidally averaged depth (m, black dots), width (m, blue dots) and cross-sectional area (m$^2$, green dots) along the Guadiana estuary. Red lines represent exponential fit curves for the width and cross-sectional area.



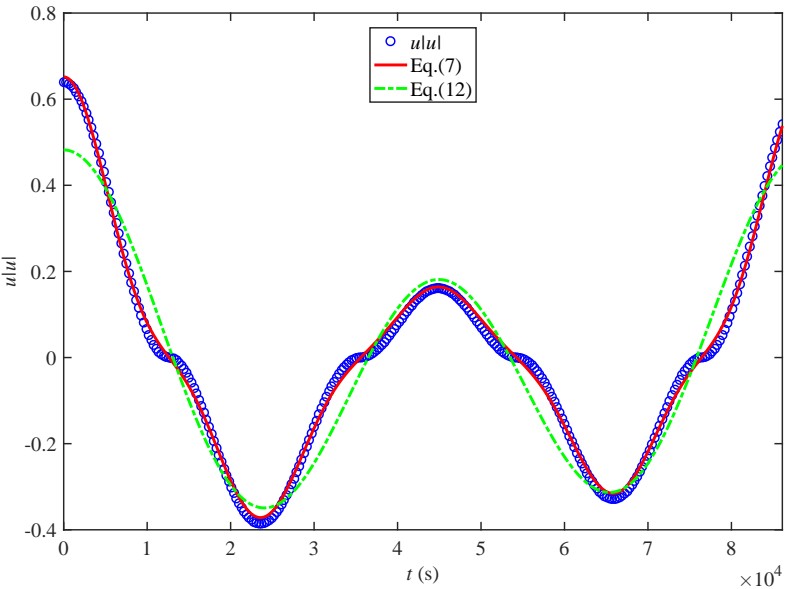

**Figure 3.** Approximation to the quadratic velocity $u|u|$ by the Chebyshev polynomials approach for the case of two tidal constituents (i.e., $M_2$ and $K_1$). Here, $u = 0.6\cos(\omega_1 t) + 0.2\cos(\omega_2 t)$, where $\omega_1$ and $\omega_2$ represent the tidal frequencies of $M_2$ and $K_1$, respectively.

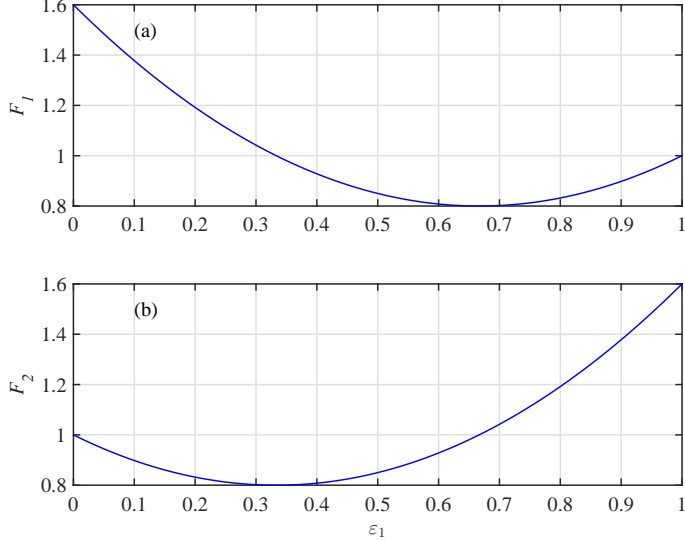

**Figure 4.** Computed effective friction coefficients $F_1$ (a) and $F_2$ (b) from Eqs. (13) and (14) as a function of $\varepsilon_1$.



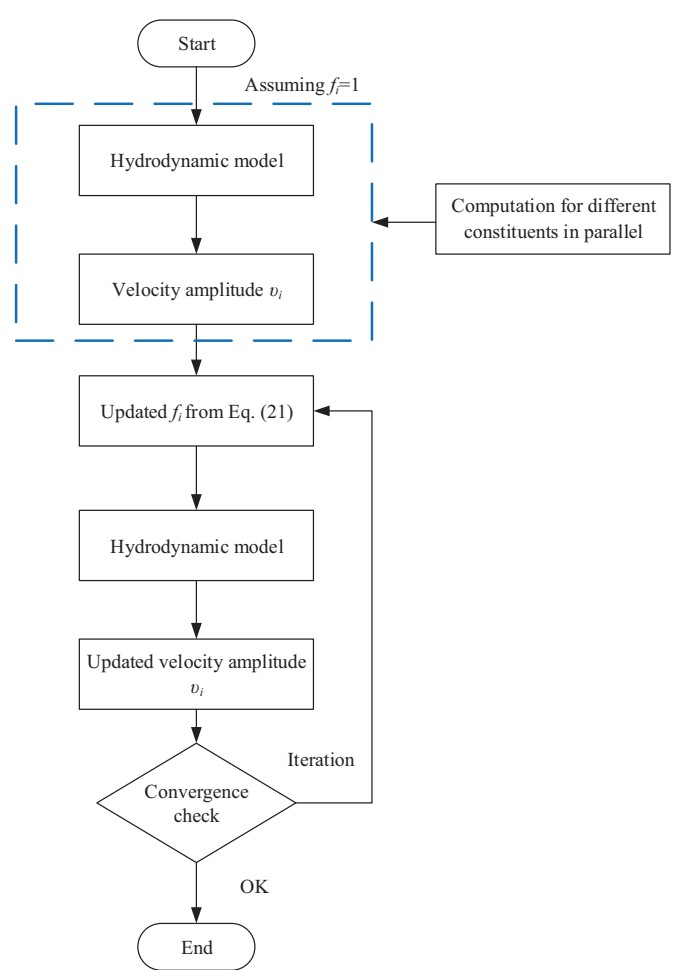

**Figure 5.** Computation process for tidal properties of different constituents in an estuary.





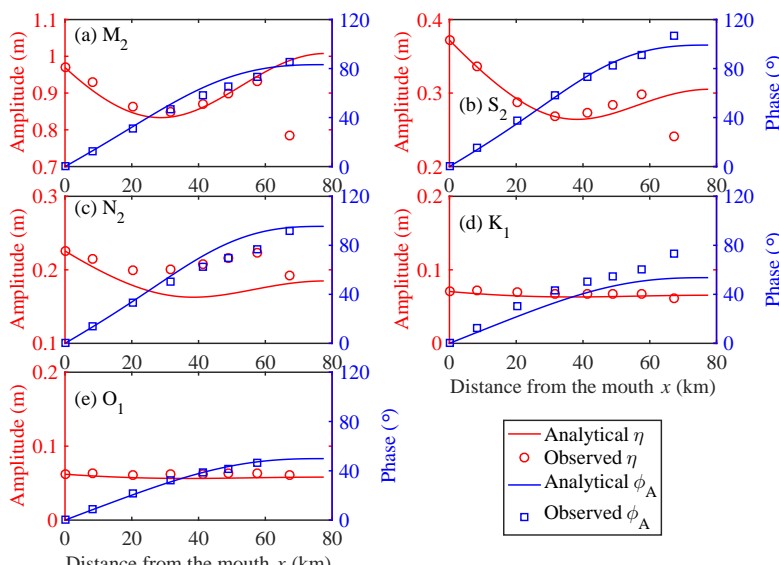

**Figure 6.** Tidal constituents (a) $M_2$; (b) $S_2$; (c) $N_2$; (d) $K_1$; (e) $O_1$: modelled against observed values of tidal amplitude (m) and phase (°) of elevation along the Guadiana estuary.

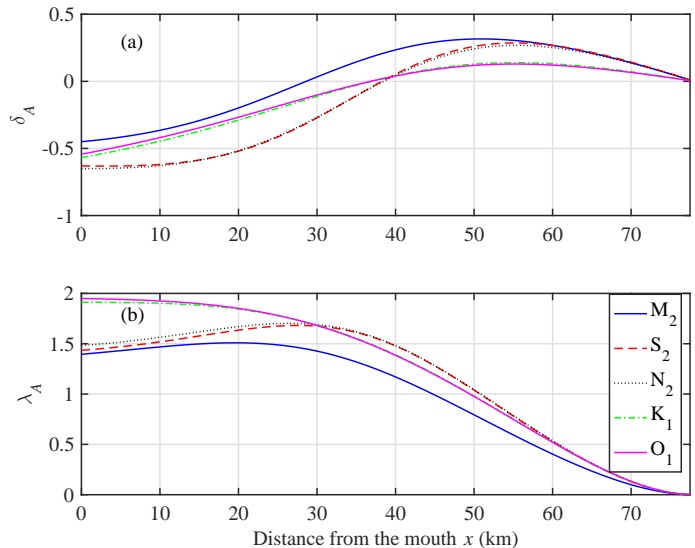

**Figure 7.** Longitudinal variations of tidal damping/amplification number $\delta_A$ (a) and wave celerity number $\lambda_A$ (b) for different tidal constituents along the Guadiana estuary.





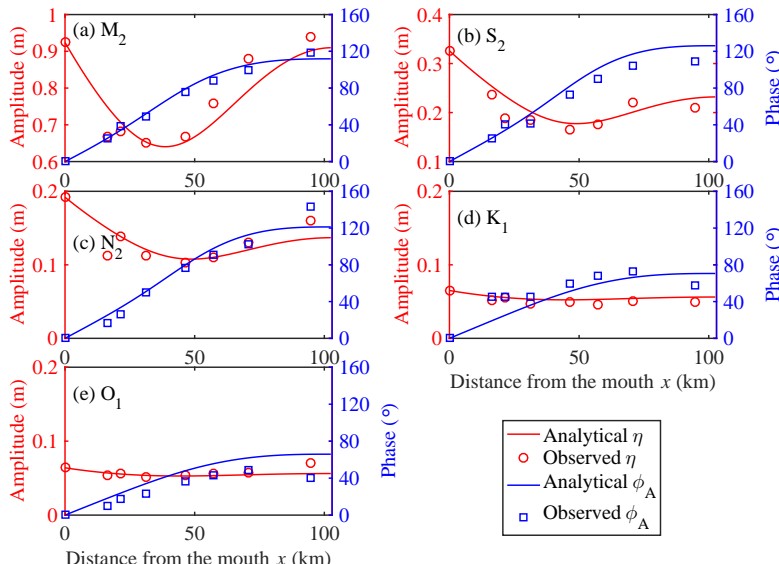

**Figure 8.** Tidal constituents (a) $M_2$; (b) $S_2$; (c) $N_2$; (d) $K_1$; (e) $O_1$: modelled against observed values of tidal amplitude (m) and phase (°) of elevation along the Guadalquivir estuary.

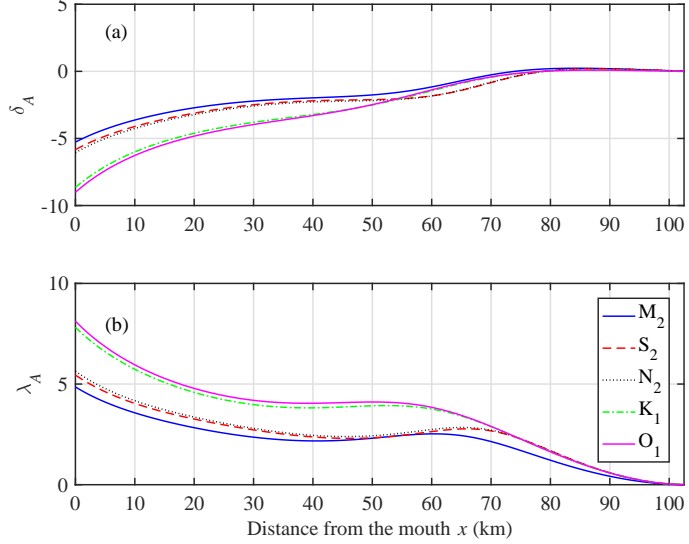

**Figure 9.** Longitudinal variations of tidal damping/amplification number $\delta_A$ (a) and wave celerity number $\lambda_A$ (b) for different tidal constituents along the Guadalquivir estuary.