# Peer review of "Frictional interactions between tidal constituents in tide-dominated estuaries"

_Ocean Science, 2018_

## Referee Comment (RC1) · J. Dronkers (Referee) · 16 May 2018

In this article an analytical solution of the tidal equations is presented to study the interaction of different tidal constituents in the Spanish Guadiana and Guadalquivir estuaries. The solution is based on a method developed by Godin (1999) and Dronkers (1964) for dealing with the non-linear friction term. Different tidal constituents derived from long-term tidal records along both estuaries are compared with amplitudes and phases of these constituents given by the analytical model. Observations and model results are in fair agreement. The article is well written and well organised.

There is probably an error in figure 9; the damping numbers (as defined in table 1) do not match the x-dependence of the amplitudes of figure 8. This is repeated in

the corresponding discussion (lines 320-327). When comparing figures 6 and 8, the damping of M2 tide in the Guadalquivir appears a bit stronger than in the Guadiana, but not an order of magnitude stronger.

The paper can be further improved by adding some clarifications concerning the following points:

1. River discharge is not mentioned at all in the paper. The influence is probably minor in the major part of the estuary, but river discharge could play a role near the sill at the upper end of the estuary, where the tidal velocities go to zero.

2. Close to the sill the tide has the appearance of a standing wave; this gives an almost infinite tidal wave celerity. Tidal wave celerity does not make much sense in this region.

3. The Chebyshev coefficients are the coefficients of the expansion of cos(nx) in powers of cos(x).

4. It should be mentioned that formula Eq. 12 gives a reasonable approximation only if the diurnal tides are much smaller than the semidiurnal tides.

5. The diurnal tides are much less damped than the semidiurnal tides. Apparently, the effects of frictional damping and channel convergence cancel approximately. This might be discussed more clearly in the paper.

6. The sensitivity of the results to the non-linear frictional interaction between the tidal constituents, being the central theme of the paper, should be discussed more explicitly. Figures 6 and 8 show the combined results of friction, channel convergence and tidal wave reflection. A figure might be added, for example, in which results with and without this frictional interaction are compared.

---

## Author Comment (AC1) · 30 May 2018

**Response letter**

In this article an analytical solution of the tidal equations is presented to study the interaction of different tidal constituents in the Spanish Guadiana and Guadalquivir estuaries. The solution is based on a method developed by Godin (1999) and Dronkers (1964) for dealing with the non-linear friction term. Different tidal constituents derived from long-term tidal records along both estuaries are compared with amplitudes and phases of these constituents given by the analytical model. Observations and model results are in fair agreement. The article is well written and well organised.

**Our reply:** We thank the Reviewer for his overall positive assessment of our work.

There is probably an error in figure 9; the damping numbers (as defined in table 1) do not match the x-dependence of the amplitudes of figure 8. This is repeated in the corresponding discussion (lines 320-327). When comparing figures 6 and 8, the damping of M2 tide in the Guadalquivir appears a bit stronger than in the Guadiana, but not an order of magnitude stronger.

**Our reply:** We thank the Reviewer for this comment. Indeed, we mixed up the unit for the tidal amplitudes imposed at the estuary mouth. The corrected Figure is displayed below (see Figure R1).

[Figure]

Figure R1. Longitudinal variations of tidal damping/amplification number $\delta_A$ (a) and wave celerity number $\lambda_A$ (b) for different tidal constituents along the Guadalquivir estuary.

In the revised paper, we shall modify the paragraph:

"*Figure 9 shows the longitudinal variations of tidal damping/amplification and wave celerity at the Guadalquivir estuary, which are similar to those in the Guadiana estuary. In general, we observe that the dominant $M_2$ tide experiences less friction than other secondary tidal constituents although it travels at more or less the same speed as other secondary tidal constituents in the seaward reach*

*(x=0~35 km). Unlike the Guadiana estuary, the damping experienced by the secondary semidiurnal tides is less than that of diurnal constituents near the estuary mouth (around x=0-7 km; Figure 9a), while the wave celerity is consistently larger in the seaward reach (x=0~38 km; Figure 9b). Similar to the Guadiana estuary, we observe that the tidal damping for the secondary semidiurnal tides is stronger than that of diurnal constituents in the central parts of the estuary (around x=7~52 km), whereas their amplifications are larger in the landward part of the estuary although wave speeds are less.*"

The paper can be further improved by adding some clarifications concerning the following points:

1. River discharge is not mentioned at all in the paper. The influence is probably minor in the major part of the estuary, but river discharge could play a role near the sill at the upper end of the estuary, where the tidal velocities go to zero.

**Our reply:** In the revised paper, we shall explicitly mention that the model does not account for the influence of river discharge on tidal wave propagation. To be more specific, in abstract part, we shall emphasize that *"The proposed method could be applicable to other alluvial estuaries with small tidal amplitude to depth ratio and negligible river discharge."* Meanwhile, in section 2.1 we shall explicitly mention that "*In order to obtain an analytical solution, we assume a negligible river discharge and that the tidal amplitude is small with respect to the mean depth and follow Toffolon and Savenije (2011) to derive the linearized solution of the system of Eqs. (3) and (4).*" In addition, in section 2.2, we shall mention that in the Guadiana estuary "*the data were collected during an extended (months-long) period of draught with negligible river discharge (e.g., always < 20 m³/s over the preceding 5 months).*", while in the Guadalquivir estuary "*the results apply to the low river discharge conditions (< 40 m³/s) that usually predominate at the estuary.*"

2. Close to the sill the tide has the appearance of a standing wave; this gives an almost infinite tidal wave celerity. Tidal wave celerity does not make much sense in this region.

**Our reply:** In the revised paper, we shall explicitly mention that: "*It is important to note that the wave celerity tends to approach infinity when tide propagates near the sill since the wave is characterized by a standing wave that is generated by the superimposition of incident and reflected waves (see also Garel and Cai, 2018).*"

3. The Chebyshev coefficients are the coefficients of the expansion of cos(nx) in powers of cos(x).

**Our reply:** We thank the Reviewer for this comment. In the revised paper, we shall clarify that "*The Chebyshev coefficients α=16/(15π) and β=32/(15π) were determined by the expansion of cos(nx) (n=1,2,…) in powers of cos(x)*".

4. It should be mentioned that formula Eq. 12 gives a reasonable approximation only if the diurnal tides are much smaller than the semidiurnal tides.

**Our reply:** In the revised paper, we shall explicitly mention this point: "*It is worth noting that Eq. (12) is a reasonable approximation only if the amplitude of the secondary constituent is much smaller than that of the dominant one*".

5. The diurnal tides are much less damped than the semidiurnal tides. Apparently, the effects of frictional damping and channel convergence cancel approximately. This might be discussed more clearly in the paper.

**Our reply:** In the revised paper, we shall include a new paragraph to clarify the difference of tidal damping between diurnal and semidiurnal tides.

*"In order to clarify the behavior of different tidal constituents, we present Fig. R2 (see below) showing the longitudinal variations of estuary shape number γ (representing the channel convergence) and friction number χ (representing the bottom friction), two major factors determining the tidal hydrodynamics, in both estuaries. Note that the variable estuary shape number γ observed in the Guadalquivir estuary is due to the adoption of a variable storage width ratio $r_S$ in the analytical model. It can be seen from Figs. 10a, c that the estuary shape numbers for diurnal tides are approximately twice larger than those for semidiurnal tides due to the tidal frequency differences (see definition of γ in Table 1). Furthermore, the effective frictions experienced by the diurnal tides are much larger than those of the semidiurnal tides due to the mutual interaction between different tidal constituents (see also Table 3). It is important to note that the propagation pattern of different tidal constituents mainly depends on the imbalance between channel convergence and friction, except for those reaches where wave reflection matters (generally close to the head). The relatively less damping experienced by diurnal tides in the seaward reach (Figures 7a and 9a) can be attributed to the fact that the channel convergence effect is much stronger than that of the semi-diurnal tides although diurnal tides experience much larger friction. In the case of the Guadalquivir estuary, we observe that the diurnal tides are more damped than those of the semidiurnal tides near the estuary mouth (x=0-7 km), which is due to the stronger bottom friction experienced by the diurnal tides. For the second (landward) half of the estuary, the less amplification experienced by diurnal tides is mainly influenced by the wave reflection from the closed end (see Garel and Cai, 2018) since both γ and χ remain more or less the same along the estuarine channels."*

[Figure]

Figure R2. Longitudinal variations of estuary shape number γ (a, c) and friction number χ (b, d) in the Guadiana estuary (a, b) and Guadalquivir estuary (c, d).

6. The sensitivity of the results to the non-linear frictional interaction between the tidal constituents, being the central theme of the paper, should be discussed more explicitly. Figures 6

and 8 show the combined results of friction, channel convergence and tidal wave reflection. A figure might be added, for example, in which results with and without this frictional interaction are compared.

**Our reply:** We thank the Reviewer for the useful suggestion. In the revised paper, we shall include a new paragraph to illustrate the importance of mutual interaction between different tidal constituents.

"*The importance of mutual interaction between different tidal constituents is illustrated with the iteratively refined model implemented at both case studies (Figures 7 and 9). For comparison, Fig. R3 (see below) shows the analytically computed damping/amplification number $\delta_A$ and celerity number $\lambda_A$ without considering mutual interaction (by setting $f_n=1$ in the model). In this case, the damping experienced by both secondary diurnal and semidiurnal tides is apparently underestimated due to the unrealistic friction adopted in the model (Fig. 11a, c). Similarly, the computed wave celerity for secondary tidal constituents is apparently overestimated due to the underestimated bottom friction. To correctly reproduce the main features of different tidal waves, it is required to use the iteratively refined model proposed in this study.*"

[Figure]

Figure R3. Longitudinal variations of damping/amplification number $\delta_A$ (a, c) and celerity number $\lambda_A$ (b, d) in the Guadiana estuary (a, b) and Guadalquivir estuary (c, d) in the absence of mutual interaction between different tidal constituents.

---

## Referee Comment (RC2) · D. Bowers (Referee) · 27 Jun 2018

review by D.G. Bowers

This paper deals with numerical modelling of several tidal constituents propagating in an estuary. This is an important problem: estuary models tend to deal with a single constituent at a time (to keep the run length down). However, the friction felt by that constituent will depend on the size and nature of the other tidal constituents in the estuary. The paper is thorough: the problem is first dealt with in an analytical way, numerical solutions are obtained and compared to observations in two estuaries in the Iberian peninsula. Agreement is good.

I'm not a numerical modeller but I know that the effect of frictional interaction between

different tidal constituents has been well studied (the important papers on the subject are referenced here). I would appreciate being told exactly what is new about this paper. Is it the first time that estuaries with an exponential shape have been considered in this way? Also, I would be interested to know if the problem could be approached just by matching model results to observations to get the best fit (as I suspect many modellers would do) without worrying too much about the theory.

The paper is well written, but it is long and technical. I don't suggest doing anything about it now, but Iw ould encourage the authors to go for a more concise style in the future. Having said that, I found myself wondering why the estuaries behave as they do. WHy does the tidal amplitude first reduce before increasing towards the tidal limit. I think I undesrtand that, but it would be interesting to read the authors opinion in the discussion section.

There were some small points I noticed which ould benefit from correction:

line 83 the storage width Bs is not defined here as far as I can see, although it is defined in the figure. At this stage I am confused about whether the model considers just a rectangular channel (with constant width) or whether the width is allowed to change with the tide.

line 115 Why would there be different celerities for elevations and velocities?

equations 10 and 11 it looks line - signs occur where there should be = signs (although that may be a trick of PDF).
* * *

---

## Author Comment (AC2) · 28 Jun 2018

**Response letter**

This paper deals with numerical modelling of several tidal constituents propagating in an estuary. This is an important problem: estuary models tend to deal with a single constituent at a time (to keep the run length down). However, the friction felt by that constituent will depend on the size and nature of the other tidal constituents in the estuary. The paper is thorough: the problem is first dealt with in an analytical way, numerical solutions are obtained and compared to observations in two estuaries in the Iberian peninsula. Agreement is good.

**Our reply:** We thank the Reviewer for his overall positive assessment of our work.

I'm not a numerical modeller but I know that the effect of frictional interaction between different tidal constituents has been well studied (the important papers on the subject are referenced here). I would appreciate being told exactly what is new about this paper. Is it the first time that estuaries with an exponential shape have been considered in this way?

**Our reply:** In the revised paper, we shall explicitly mention that "*Unlike the previous studies exploring the effect of frictional interaction between different tidal constituents by quantifying a friction correction factor only (e.g., Dronkers, 1964; Le Provost, 1973; Pingree, 1982; Fang, 1987; Godin, 1999; Inoue and Garrett, 2007), in this study, for the first time, the mutual interactions between tidal constituents in the frictional term were explored using **a conceptual model** by means of expanding the quadratic velocity using a Chebyshev polynomials approach which allows for defining a friction correction factor for each constituent.*" The advantage of such conceptual model lies in the deterministic description of the mutual frictional interaction among tidal constituents, which avoids the need of an independent calibration of the friction parameter for the single constituent. The proposed method can be used as a prognostic tool to study the propagation of different tidal constituents in convergent estuaries where the cross-sectional area can be described by an exponential function.

Also, I would be interested to know if the problem could be approached just by matching model results to observations to get the best fit (as I suspect many modellers would do) without worrying too much about the theory.

**Our reply:** Exactly! Similar to our previous analytical studies for a single tidal constituent (e.g., Toffolon and Savenije, 2011; Cai et al., 2016), the implementation of the new model accounting for the nonlinear interactions between tidal constituents also requires a few dimensionless input parameters representing the external tidal forcing and estuary geometry, which are independent of the tidal hydrodynamics along the estuary. Hence, the problem does solve by matching the model results to observations.

The paper is well written, but it is long and technical. I don't suggest doing anything about it now, but I would encourage the authors to go for a more concise style in the future. Having said that, I found myself wondering why the estuaries behave as they do. WHy does the tidal amplitude first reduce before increasing towards the tidal limit. I think I undesrtand that, but it would be interesting to read the authors opinion in the discussion section.

**Our reply:** We thank the Reviewer for the useful suggestion. In the revised paper, we shall explicitly mention the underlying mechanism of tidal hydrodynamics (i.e., damping/amplification along the channel) in these two estuaries. In particular, the tidal damping along the first half of the estuaries is mainly due to the damping of the dominant $M_2$ wave owning to the fact that the impact of bottom friction dominates over the channel convergence. Along the upper reach, enhanced morphological convergence and reflection effects (that reduce the overall friction experienced by the propagating wave) result in the overall amplification of the tidal wave. For more details of the tidal hydrodynamics in these two estuaries, readers can refer to Garel and Cai (2018) for the Guadiana estuary and Diez-Minguito et al. (2012) for the Guadalquivir estuary.

There were some small points I noticed which could benefit from correction:
line 83 the storage width Bs is not defined here as far as I can see, although it is defined in the figure. At this stage I am confused about whether the model considers just a rectangular channel (with constant width) or whether the width is allowed to change with the tide.

**Our reply:** In the revised paper, we shall explicitly define the storage width $B_S$ as "***width of the channel at averaged high water level***". In this study, we assume a rectangular cross-section with a constant width since the variation of width $\Delta\bar{B}$ with time is usually negligible (i.e., $\Delta\bar{B}/B \ll 1$). On the other hand, the overall influence of storage area is represented by the storage width ratio, defined as the ratio of the storage width $B_S$ (width of the channel at averaged high water level) to the tidally averaged width (i.e., $r_S = B_S/\bar{B}$).

line 115 Why would there be different celerities for elevations and velocities?

**Our reply:** It was shown by Savenije et al. (2008) that for an infinitely long channel the wave celerities for elevation and velocity are almost the same due to the combined impacts of bottom friction and channel convergence. However, for a semi-closed channel the wave celerities for elevation and velocity would deviate due to the additional impact of reflected wave at the closed end (e.g., Toffolon and Savenije, 2011). Such a celerity difference was recently investigated and detailed by Garel and Cai (2018) for the case of the Guadiana estuary.

equations 10 and 11 it looks line - signs occur where there should be = signs (although that may be a trick of PDF).

**Our reply:** This is probably due to the PDF viewer, as there is not such typing error on our version.

**References**

Cai, H., Toffolon, M., and Savenije, H. H. G.: An Analytical Approach to Determining Resonance in Semi-Closed Convergent Tidal Channels, Coast Eng. J., 58, doi:Artn 1650009 10.1142/S0578563416500091, 2016.

Diez-Minguito, M., Baquerizo, A., Ortega-Sanchez, M., Navarro, G., and Losada, M. A.: Tide transformation in the Guadalquivir estuary (SW Spain) and process-based zonation, J. Geophys. Res., 117, doi:10.1029/2011jc007344, 2012.

Dronkers, J. J.: Tidal computations in River and Coastal Waters, Elsevier, New York, 1964.

Fang, G.: Nonlinear effects of tidal friction, Acta Oceanol. Sin., 6 (Suppl.), 105–122, 1987.

Garel, E. and Cai, H.: Effects of Tidal-Forcing Variations on Tidal Properties Along a Narrow Convergent Estuary, Estuar. Coast, doi: 10.1007/s12237-018-0410-y, 2018.

Godin, G.: The propagation of tides up rivers with special considerations on the upper Saint Lawrence river, Estuar. Coast. Shelf S., 48, 307–324, doi:10.1006/ecss.1998.0422, 1999.

Inoue, R. and Garrett, C.: Fourier representation of quadratic friction, J. Phys. Oceanogr., 37, 593–610, doi:10.1175/Jpo2999.1, 2007.

Le Provost, C.: D´ecomposition spectrale du terme quadratique de frottement dans les ´equations des mar´ees littorales, C.R. Acad. Sci. Paris, 276, 653–656, 1973.

Pingree, R. D.: Spring Tides and Quadratic Friction, Deep-Sea Res. Part A-Oceanographic Research Papers, 30, 929–944, doi:10.1016/0198-0149(83)90049-3, 1983.

Savenije, H. H. G., Toffolon, M., Haas, J., and Veling, E. J. M.: Analytical description of tidal dynamics in convergent estuaries, J. Geophys. Res., 113, doi:10.1029/2007JC004408, 2008.

Toffolon, M. and Savenije, H. H. G.: Revisiting linearized one-dimensional tidal propagation, J. Geophys. Res., 116, doi:10.1029/2010JC006616, 2011.

---

## Author Response (AR1)

**Response letter**

We thank the Editor and the Reviewers for the careful consideration of our work. In the revised paper, we have addressed all the comments formulated by the Reviewers by replying (in black) to their remarks (in blue). The lines numbers in this rebuttal refer to the revised version of the manuscript.

**Responses to comments by Reviewer #1**

In this article an analytical solution of the tidal equations is presented to study the interaction of different tidal constituents in the Spanish Guadiana and Guadalquivir estuaries. The solution is based on a method developed by Godin (1999) and Dronkers (1964) for dealing with the non-linear friction term. Different tidal constituents derived from long-term tidal records along both estuaries are compared with amplitudes and phases of these constituents given by the analytical model. Observations and model results are in fair agreement. The article is well written and well organised.

Our reply: We thank the Reviewer for his overall positive assessment of our work.

There is probably an error in figure 9; the damping numbers (as defined in table 1) do not match the x-dependence of the amplitudes of figure 8. This is repeated in the corresponding discussion (lines 320-327). When comparing figures 6 and 8, the damping of M2 tide in the Guadalquivir appears a bit stronger than in the Guadiana, but not an order of magnitude stronger.

**Our reply:** We thank the Reviewer for this comment. Indeed, we mixed up the unit for the tidal amplitudes imposed at the estuary mouth. The corrected Figure is displayed below (see Figure R1).

Figure R1. Longitudinal variations of tidal damping/amplification number  $\delta_A$  (a) and wave celerity number  $\lambda_A$  (b) for different tidal constituents along the Guadalquivir estuary.

In the revised paper, we have modified the paragraph as follows:

"Figure 9 shows the longitudinal variations of tidal damping/amplification and wave celerity for the Guadalquivir estuary, which are similar to those in the Guadiana estuary. In general, we observe that the dominant  $M_2$  tide experiences less friction than other secondary semidiurnal tidal constituents although it travels at more or less the same speed in the seaward reach (x=0-35 km). Unlike the Guadiana estuary, the damping experienced by the secondary semidiurnal tides is less than those of diurnal constituents near the estuary mouth (around x=0-7 km; Figure 9a), while the wave celerity is consistently larger in the seaward reach (x=0-38 km; Figure 9b). Similar to the Guadiana estuary, we observe that the tidal damping for the secondary semidiurnal tides is stronger than those of diurnal constituents in the central parts of the estuary (around x=7-52 km), whereas their amplifications are larger in the landward part of the estuary although their wave speeds are less." (see lines 333-341)

The paper can be further improved by adding some clarifications concerning the following points:

1. River discharge is not mentioned at all in the paper. The influence is probably minor in the major part of the estuary, but river discharge could play a role near the sill at the upper end of the estuary, where the tidal velocities go to zero.

**Our reply:** In the revised paper, we have explicitly mentioned that the model does not account for the influence of river discharge on tidal wave propagation. To be more specific, in abstract part, we emphasized that *"The proposed method could be applicable to other alluvial estuaries with small tidal amplitude to depth ratio and negligible river discharge."* (see lines 13-14)

Meanwhile, in section 2.1 we have explicitly mentioned that "In order to obtain an analytical solution, we assume a negligible river discharge and that the tidal amplitude is small with respect to the mean depth and follow Toffolon and Savenije (2011) to derive the linearized solution of the system of Eqs. (3) and (4)." (see lines 95-96)

In addition, in section 2.2, we have explicitly mentioned that in the Guadiana estuary "the data were collected during an extended (months-long) period of drought with negligible river discharge (e.g., always < 20 m3/s over the preceding 5 months)." (see lines 149-151), while in the Guadalquivir estuary "the results apply to the low river discharge conditions (< 40 m3/s) that usually predominate at the estuary." (see lines 168-169)

2. Close to the sill the tide has the appearance of a standing wave; this gives an almost infinite tidal wave celerity. Tidal wave celerity does not make much sense in this region.

**Our reply:** In the revised paper, we have explicitly mentioned that: "*It is important to note that a standing wave pattern with celerity approaching infinity is produced near the sill due to the superimposition of the incident and reflected waves (see also Garel and Cai, 2018)."* (see lines 322-324)

3. The Chebyshev coefficients are the coefficients of the expansion of cos(nx) in powers of cos(x).

**Our reply:** We thank the Reviewer for this comment. In the revised paper, we have clarified that "*The Chebyshev coefficients*  $\alpha = 16/(15\pi)$  and  $\beta = 32/(15\pi)$  were determined by the expansion of cos(nx) (n=1,2,...) in powers of cos(x) (Godin, 1991, 1999)". (see lines 176-178)

4. It should be mentioned that formula Eq. 12 gives a reasonable approximation only if the diurnal tides are much smaller than the semidiurnal tides.

**Our reply:** In the revised paper, we have explicitly mentioned this point: "*It is worth noting that Eq. (12) is a reasonable approximation only if the amplitude of secondary constituent is much smaller than that of the dominant one*". (see lines 205-206)

5. The diurnal tides are much less damped than the semidiurnal tides. Apparently, the effects of frictional damping and channel convergence cancel approximately. This might be discussed more clearly in the paper.

**Our reply:** In the revised paper, we have included a new paragraph to clarify the difference of tidal damping between diurnal and semidiurnal tides.

"In order to clarify the behavior of different tidal constituents, we present Figure 10 [see Figure R2 below] showing the longitudinal variations of estuary shape number  $\gamma$  (representing the channel convergence) and friction number  $\chi_n$  (representing the bottom friction), two major factors determining the tidal hydrodynamics, in both estuaries. Note that the variable estuary shape number  $\gamma$  observed in the Guadalquivir estuary is due to the adoption of a variable storage width ratio  $r_s$  in the analytical model. On the one hand, the estuary shape numbers for diurnal tides are approximately twice larger than those for semidiurnal tides (Figures 10a, d) due to the tidal frequency differences (see definition of  $\gamma$  in Table 1). On the other hand, the effective friction experienced by the diurnal tides is much larger than those of the semidiurnal tides due to the mutual interaction between different tidal constituents (Figure 10b, e, see also Table 3). However, the propagation of different tidal constituents mainly depends on the imbalance between channel convergence and friction, except for those reaches where wave reflection matters (generally close to the head). In particular, in the seaward reach the tidal damping for each tidal constituent can be approximately estimated by  $\delta_A = \gamma/2 - \chi_n \mu \cos(\phi)/(2\lambda_A)$  (see equation (20) by Cai et al., 2012). While the channel convergence effect (represented by  $\gamma/2$ ) is much stronger for diurnal tides than for semidiurnal tides, the frictional effect (represented by  $\chi_n \mu \cos(\phi)/(2\lambda_A)$ ) is only slightly larger (Figure 10c, f). Hence, diurnal tides experience a relatively lower damping in the

seaward reach (Figures 7a and 9a). For instance, in the case of the Guadalquivir estuary, diurnal tides are more damped than semidiurnal tides near the estuary mouth (x=0-7 km). For the second (landward) half of the estuary, the lower amplification experienced by diurnal tides is mainly due to the wave reflection from the closed end (see Garel and Cai, 2018)." (see lines 349-369)

Figure R2. Longitudinal variations of estuary shape number  $\gamma$  (a, d), friction number  $\chi_n$  (b, e) and  $\chi_n \mu \cos(\phi)/(2\lambda_A)$  (c, f) in the Guadiana estuary (a, b, c) and Guadalquivir estuary (d, e, f).

The sensitivity of the results to the non-linear frictional interaction between the tidal constituents, being the central theme of the paper, should be discussed more explicitly. Figures 6 and 8 show the combined results of friction, channel convergence and tidal wave reflection. A figure might be added, for example, in which results with and without this frictional interaction are compared.

**Our reply:** We thank the Reviewer for the useful suggestion. In the revised paper, we have included a new paragraph to illustrate the importance of mutual interaction between different tidal constituents:

"The importance of mutual interaction between different tidal constituents is illustrated with the iteratively refined model implemented at both case studies (Figures 7 and 9). For comparison, Figure 11 [see Figure R3 below] shows the analytically computed damping/amplification number  $\delta_A$  and celerity number  $\lambda_A$  without considering mutual interaction (by setting  $f_n=1$  in the model). In this case, the damping experienced by both secondary diurnal and semidiurnal tides are apparently underestimated due to the unrealistic friction adopted in the model (Figure 11a, c, see also Figures 7a and 9a, respectively). Similarly, the computed wave celerity for secondary tidal

constituents are apparently overestimated due to the underestimated bottom friction (Figure 11b, d, see also Figures 7b and 9b, respectively). To correctly reproduce the main features of different tidal waves, it is required to use the iteratively refined model proposed in this study." (see lines 370-379)